# Scientific Opinion Summarization: Paper Meta-review Generation Dataset, Methods, and Evaluation

**Author Name**
email@example.com

## Abstract

Opinions in scientific research papers can be divergent, leading to controversy or consensus among reviewers. However, most existing datasets for opinion summarization are centered around product reviews and assume that the analyzed opinions are non-controversial, failing to account for the variability seen in other contexts such as academic papers, political debates, or social media discussions. To address this gap, we propose the task of scientific opinion summarization, where research paper reviews are synthesized into meta-reviews. To facilitate this task, we introduce the ORSUM dataset covering 15,062 paper meta-reviews and 57,536 paper reviews from 47 conferences. Furthermore, we propose the Checklist-guided Iterative Introspection (CGI$^2$) approach, which breaks down scientific opinion summarization into several stages, iteratively refining the summary under the guidance of questions from a checklist. Our experiments show that (1) human-written summaries do not always satisfy all necessary criteria such as depth of discussion, and identifying consensus and controversy for the specific domain, and (2) the combination of task decomposition and iterative self-refinement shows strong potential for enhancing the opinions and can be applied to other complex text generation using black-box LLMs.

## 1 Introduction

Opinion Summarization traditionally targets product reviews, aiming to distill representative opinions on key product aspects such as product quality and price. This assumes a dominant, singular opinion within the texts being summarized [Hu and Liu, 2006; Amplayo *et al.*, 2021b; Angelidis and Lapata, 2018; Suhara *et al.*, 2020]. However, this approach often overlooks the nuanced and multi-faceted nature of discussions in scientific documents, where multiple viewpoints may coexist and no single opinion dominates.

Furthermore, most opinion summarization datasets in the product domain for abstractive summarization are synthetic, containing redundant cut-and-paste extracts built by combining extracted snippets, or by sampling a review from the

Figure 1: Product meta-reviews and paper meta-reviews have different compositions: A product meta-review presents the most prominent opinion instead of summarizing opinions, while a paper meta-review summarizes different opinions and makes recommendations.

collection and pretending that it is a gold-standard meta-review [Amplayo *et al.*, 2021b].

To address this gap, we introduce the new task of **Scientific Opinion Summarization**, where a set of opinions must be synthesized into a meta-opinion that justifies a decision. Scientific Opinion Summarization aims to provide a succinct synopsis for scientific documents, helping readers to recap salient information and understand the professional discussion. Scientific meta-reviews, in particular, summarize the *controversies* and *consensuses* in the reviews, guiding decision making such as the acceptance or rejection of a paper. Taking research paper meta-review generation as a typical scenario, we build the **ORSUM** dataset by collecting open-sourced paper and meta-reviews from OpenReview[1], covering 15,062 meta-reviews and 57,536 reviews from 47 conference venues. Compared to synthetic datasets from product review domains, ORSUM is built upon large-scale real-world data, enabling applications of supervised abstractive summarization methods and more fine-grained tex-

---

[1] https://openreview.net/

tual analysis. In addition to meta-review generation, OR-SUM's structured content, including ratings on different aspects such as if agreements/disagreements are present alongside strengths/weaknesses and multi-turn discussions, will benefit a wide range of related tasks, such as review generation [Wang *et al.*, 2020], recommendation prediction [Deng *et al.*, 2020; Friedl *et al.*, 2021], review rating prediction [Li *et al.*, 2017; Chan *et al.*, 2020], and argument pair extraction [Cheng *et al.*, 2020].

The task of Scientific Opinion Summarization presents a distinct set of challenges, including (1) *Decision Consistency*: Whether the Meta-review aligns with the decision, which guides opinion selection and discussion in the meta-review. Generated scientific meta-reviews should reflect these decisions. (2) *Discussion involvement*: Unlike product meta-reviews that rely on majority voting, scientific meta-reviews assess both the pros and cons, as well as opinion agreement and disagreement, to evaluate the paper from the perspective of a more senior reviewer.

To tackle these challenges, we propose Checklist-guided Iterative Introspection ($CGI^2$). $CGI^2$ first breaks the task of scientific opinion summarization into multiple steps, constantly requesting evidence to mitigate both LLMs' inability to follow complicated instructions and their tendency to produce hallucinations. To enhance discussion involvement, $CGI^2$ iteratively revises the generated meta-review based on a predefined checklist. Finally, we identify key aspects a meta review should satisfy to be of high quality, and propose ways to evaluate these aspects using reference-free LLM-based metrics.

Our contributions include the following:

- We introduce the task of scientific opinion summarization and construct the ORSUM dataset, which contains 15,062 meta-reviews and 57,536 reviews from 47 conferences on OpenReview. It is currently the largest paper meta-review dataset.

- We propose Checklist-guided Iterative Introspection ($CGI^2$), which breaks down the task of scientific opinion summarization into several stages and iteratively refines the summary under the guidance of questions from a checklist.

- We construct a comprehensive evaluation framework for meta-review generation and assess the different summarization paradigms on ORSUM.

## 2 Related Work

### 2.1 Opinion Summarization

The task of opinion summarization is typically decomposed into three stages: aspect extraction, which identifies the specific features discussed in reviews; polarity identification, which assesses whether the sentiment towards each aspect is positive, negative, or neutral; and summary generation, which compiles these aspects and sentiments into a cohesive summary of the opinions [Hu and Liu, 2006]. The lack of parallel data in review summaries limits most methodologies into the few-shot abstractive setting [Brazinskas *et al.*, 2020a;

Brazinskas *et al.*, 2022], or unsupervised extractive setting [Angelidis and Lapata, 2018; Angelidis *et al.*, 2020; Chowdhury *et al.*, 2022] where the aspects and sentiments from the input reviews are collected, selected, and rearranged into the output meta-reviews.

Only a few previous opinion summarization datasets [Wang and Ling, 2016] contain gold-standard summaries and support supervised training of abstractive models [Amplayo and Lapata, 2019]. Pretrained aspect-based sentiment analysis [Suhara *et al.*, 2020], variational autoencoders [Brazinskas *et al.*, 2020b; Chu and Liu, 2019; Iso *et al.*, 2021; Isonuma *et al.*, 2021] and large language models [Bhaskar *et al.*, 2022] enable unsupervised abstractive approaches, where the generated summaries are validated to be more fluent, informative, coherent, and concise compared to traditional extractive summaries.

To support the training and evaluation of supervised methods, recent work constructs synthetic datasets by random sampling [Shen *et al.*, 2023], adding noise to the sampled summary to generate documents [Amplayo and Lapata, 2020], searching for relevant reviews to act as the input document set [Elsahar *et al.*, 2021], or sampling with trained models [Amplayo *et al.*, 2021a; Amplayo *et al.*, 2021b]. However, synthetic pseudo-summaries in the product review domain are known to be detached from real-world distributions, be possibly irrelevant or inconsistent with input documents, and are known to ignore important underlying details.

### 2.2 Meta-review Generation

The first attempt to generate paper meta-reviews is Meta-Gen [Bhatia *et al.*, 2020], which generates an extractive summary draft then uses a fine-tuned model for decision prediction and abstractive review generation. [Kumar *et al.*, 2021] emphasizes decision awareness, proposing a model for decision prediction and subsequent meta-review generation. The most similar work to ours is MReD [Shen *et al.*, 2022], where 7,089 paper meta-reviews from ICLR 2018 - 2021 are manually annotated with sentence-level structure labels. These structure labels categorize sentences based on their function in the document, such as summary, evaluation, or recommendation. The difference between their work and ours is that they focus on structure-controlled text generation, while our work 1) enables scientific opinion summarization with a larger corpus, 2) provides a prompting-based solution, and 3) performs broader evaluations. Note that while there are other concurrent efforts to collect paper meta-reviews or reviews [Dycke *et al.*, 2023], we are the first to model meta-review generation as scientific opinion summarization and to offer a unified dataset covering a broad range of conference venues.

## 3 Task Formulation

Given a research paper's title, abstract, and set of reviews, the goal of **Scientific Opinion Summarization** is to generate a meta-review summarizing the reviews' opinions in order to make a decision recommendation for acceptance or rejection.

As noted by ACL's area chair guidance[2], meta-reviews

---

[2]https://aclrollingreview.org/aetutorial

| Dataset | Collection | Count(SRC) | Count(TRG) | Len(SRC) | Len(TRG) | Novel 4-gram | NID |
|---|---|---|---|---|---|---|---|
| RT | Human | 246,164 | 3,731 | 20.57 | 21.4 | 97.10 | 0.1615 |
| Copycat | AMT | 480 | 180 | 42.63 | 54.33 | 89.62 | 0.2506 |
| OPOSUM | AMT | 600 | 60 | 43.51 | 67.77 | 85.92 | 0.1260 |
| Yelp | AMT | 3,200 | 200 | 65.25 | 61.15 | 93.26 | 0.1661 |
| DENOISESUM | Synthetic | 73282 | 837 | 24.32 | 26.45 | 94.12 | 0.2270 |
| PLANSUM | Synthetic | 249,844 | 869 | 42.81 | 97.2 | 91.40 | 0.2395 |
| SPACE | Human | 5000 | 1050 | 34.27 | 54.38 | 90.38 | 0.1671 |
| **ORSUM** | Human | 57,536 | 15,062 | 376.36 | 141.76 | 99.89 | 0.1572 |

Table 1: We compare ORSUM with existing opinion summarization datasets that contain gold-standard summaries. SRC refers to the source or input reviews. TRG refers to the target or output meta-reviews. A higher novel 4-gram score suggests better abstractiveness, while a lower NID score implies less redundancy.

summarize reviews by aggregating opinions to support the decision. The task entails summarizing the paper's key strengths and weaknesses and explicitly evaluating whether those strengths surpass the weaknesses.

## 4 ORSUM Dataset

### 4.1 Dataset Collection and Preprocessing

To facilitate the task of scientific opinion summarization, we collect the **ORSUM** dataset which consists of human-written meta-reviews from OpenReview. The dataset contains each paper's URL, title, abstract, decision, meta-review from the area chair, and reviews from individual reviewers. We crawl 15,062 paper meta-reviews and 57,536 individual reviews from 47 conference venues. Papers with meta-reviews shorter than 20 tokens and comments made by non-official reviewers are excluded. The data format is unified across venues, and we provide train/validation/test splits with 9,890/549/550 samples for convenient usage by future works.

### 4.2 Dataset Comparison

We compare ORSUM with existing opinion summarization datasets (or their subsets) with gold-standard summaries, including The Rotten Tomatoes (RT) [Wang and Ling, 2016], Copycat [Brazinskas *et al.*, 2020b], OPOSUM [Angelidis and Lapata, 2018], Yelp [Chu and Liu, 2019], DENOIS-ESUM [Amplayo and Lapata, 2020], PLANSUM [Amplayo *et al.*, 2021b], and SPACE [Angelidis *et al.*, 2021] datasets. To perform a quantitative comparison, we utilize two key metrics:

**Abstractiveness.** The percentage of novel n-grams in a meta-review is defined by the ratio of n-grams which do not appear in the source reviews, to the total number of n-grams in the meta review. This metric intuitively measures the abstractiveness of the summaries [Chen *et al.*, 2021]. Table 1 indicates a greater degree of abstractiveness in ORSUM.

**Redundancy.** To examine the presence of insightful information in the input reviews, we assess redundancy using the Normalized Inverse of Diversity (NID) score [Xiao and Carenini, 2020] This score is calculated as the inverse of the diversity metric, which measures the variability of information in the reviews with length normalization: $NID = 1 - \frac{(entropy(D))}{log(|D|)}$. A higher NID signifies greater redundancy. Table 1 shows lower redundancy in ORSUM, which can be

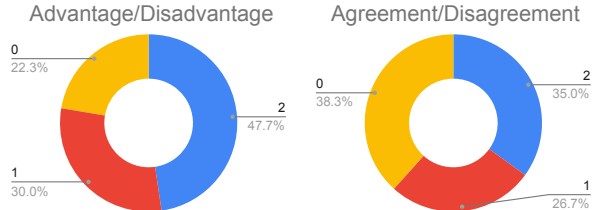

Figure 2: Meta-review composition. The scores range from 0 to 2: 0 indicates that the meta-review does not address the discussion at all. 1 signifies that the meta-review incorporates the discussion but lacks concrete evidence. 2 denotes that the meta-review involves a detailed discussion. Only 47.7% and 35.0% of meta-reviews meet the fundamental criteria for discussions of advantages and disadvantages, and consensus and controversy, respectively.

attributed to the fact that many reviews address distinct aspects of their papers.

### 4.3 Composition Analysis

To investigate whether ORSUM's human-authored meta-reviews discuss both a paper's pros/cons and the reviews' level of agreement/disagreement, we conduct a human evaluation focused on meta-review composition. Three annotators are asked to assess the meta-reviews in terms of **discussion involvement**: how effectively a summary engages with the content by discussing the paper's advantages/disadvantages, and by discussing the agreements/disagreements of the reviews. Annotation scores range from 0 (no involvement) to 2 (detailed involvement).

Our evaluation results depicted in Figure 2 reveal that only 20.7% of meta-reviews include an assessment of both advantages/disadvantages and review agreements/disagreements, regardless of their length. For each category, 47.7%, and 35.0% of meta-reviews meet the criteria of containing discussions of advantages and disadvantages and discussions of agreements/disagreements, respectively. Based on these results, we conclude that *human-written meta-reviews do not always meet the necessary criteria for an effective meta review, and they may be unsuitable for developing summarization models as supervised training signals. The low percentage of comprehensive reviews highlights a gap in coverage and thoroughness that can affect the performance and reliability of models trained on these summaries.*

# 5 Checklist-guided Iterative Introspection Method for Meta-review Generation

Motivated by the unreliability of human-written meta-reviews, we turn to Large Language Models (LLMs) like ChatGPT [OpenAI, 2021] for meta-review generation. We choose LLMs for their world knowledge, and their potential to generate reviews efficiently and scalably. However, LLMs struggle to follow complicated instructions, and have a tendency to produce hallucinations. To mitigate these deficiencies, we propose to break the task of scientific review generation into multiple steps, consistently requesting evidence for each step. To enhance discussion involvement and evidence-based coherence in the generation process, we further introduce a checklist-guided self-feedback mechanism. Our method is similar to the process of self-refinement [Madaan *et al.*, 2023], which involves the LLM iteratively revising the generated meta-review based on its own feedback. Unlike prior work, however, our checklist-guided self-feedback uses self-feedback derived from questions in a predefined checklist, ensuring that the revision process progresses towards our desired criteria. Figure 3 illustrates our proposed Checklist-guided Iterative Introspection ($CGI^2$) method.

**Initial Run.** Given a paper's title, abstract, and set of reviews, $CGI^2$ generates a draft of the meta-review in four steps: (1) For each review, we prompt the LLM to extract and rank opinions, while including sentiment, aspect, and evidence. Due to the input length constraint, each review is truncated to 300 tokens. (2) Based on the extracted opinions, we prompt the LLM to list the paper's most important advantages and disadvantages, the evidence for those statements, and those statements' corresponding reviewers. (3) We prompt the LLM to list the consensuses and controversies in the reviews, the evidence for those statements, and their corresponding reviewers. (4) Given the paper's acceptance or rejection decision, we prompt the LLM to write a meta-review based on the information extracted in steps (1)–(3).

**Iterative Runs.** With the meta-review draft from the initial four-step run, $CGI^2$ iteratively poses questions, obtains self-feedback, and requests further refinement. In each run, we first select an assessment question from a pre-constructed list of questions, as shown in Table 2. This checklist, customized for meta-review generation, covers the four most crucial aspects of meta-reviews. The checklist can also easily be expanded and adapted to other complex text generation tasks. After prompting the LLM with the assessment questions, we collect the refinement suggestions from the LLM's. These refinement suggestions are used as prompts to generate a revised version of the meta-review. The checklist questions are posed sequentially in one iterative run, with the number of iterations set as a hyper-parameter in $CGI^2$.

Our proposed approach offers two key benefits. First, it eliminates the need for external scoring functions that demand training data or human annotations. Second, it provides a general solution for employing LLMs as black boxes in complex text generation tasks.

# 6 Evaluation

Meta-review generation requires a system to accurately summarize opinions, highlight reviewer consensuses and controversies, offer judgments, and make recommendations. The task's complexity thus requires an evaluation that is multifaceted and goes beyond n-gram similarity. However, current evaluation metrics for long text generation are inadequate to measure the particular requirements of meta-review generation. To address this gap, we propose a comprehensive evaluation framework that combines standard evaluation metrics with LLM-based evaluation metrics.

## 6.1 Standard Metrics

We apply standard metrics in natural language generation to assess *relevance*, *factual consistency*, and *semantic coherence*. For relevance, ROUGE-L [Lin, 2004] quantifies the similarity between the generated and reference texts by calculating the longest common subsequence, while BERTScore [Zhang *et al.*, 2020] offers a more nuanced relevance evaluation by leveraging contextualized embeddings without relying on n-gram overlaps. For factual consistency, FACTCC [Kryscinski *et al.*, 2019] checks whether a given claim in the generated text is consistent with the facts presented in the source document, while SummaC [Laban *et al.*, 2021] utilizes sentence-level natural language inference models for inconsistency detection. For semantic coherence, DiscoScore [Zhao *et al.*, 2022] presents six BERT-based model variants to measure discourse coherence. We average the scores from these six models as the coherence indicator. The references used in our reference-free evaluation metrics are sourced from a test subset of our dataset, where the instances are chosen for their relevance and quality. These references provide a practical benchmark that mirrors current standards in meta-review generation at top conferences.

## 6.2 LLM-based Metrics

The aforementioned methods do not evaluate discussion involvement or evidence-decision consistency. Some reference summaries may not include discussions or utilize evidence to substantiate decisions. To address this, we propose supplementary measures for this task that can be assessed and quantified using reference-free LLM-based metrics. We aim to assess the following key aspects:

- Discussion involvement: whether the meta-review discusses the paper's strengths and weaknesses, and the paper's agreements and disagreements amongst reviewers.

- Opinion Faithfulness: whether the meta-review contradicts reviewers' opinions.

- Decision Consistency: whether the meta-review accurately reflects the final decision.

Despite its prevalence, the GPTScore [Fu *et al.*, 2023] metric requires its criteria to be described as a single word, a requirement incompatible with our detailed criteria. On the other hand, G-EVAL [Liu *et al.*, 2023] assesses the quality of NLG outputs by utilizing chain-of-thought (CoT) and a form-filling paradigm. It has been shown to have a very high correlation with human-based judgments. G-EVAL uses carefully

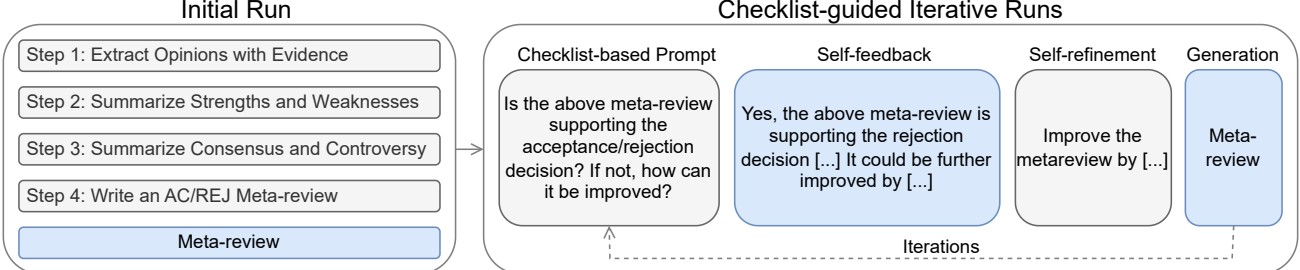

Figure 3: Our proposed CGI$^2$ framework operates through multiple iterations. In the initial iteration, the task is divided into four steps: (1) Review Opinion Extraction, (2) Strength and Weakness Synthesis, (3) Consensus and Controversy Analysis, and (4) Meta-review Drafting. For subsequent iterations, we present the black-box LLM with a query from a predefined list, acquire self-feedback, and request additional refinements.

---

1. Are the most important advantages and disadvantages discussed in the above meta-review? If not, how can it be improved?
2. Are the most important consensus and controversy discussed in the above meta-review? If not, how can it be improved?
3. Is the above meta-review contradicting reviewers' comments? If so, how can it be improved?
4. Is the above meta-review supporting the acceptance/rejection decision? If not, how can it be improved?

---

Table 2: The extensible and easily adaptable checklist for Meta-review Generation accesses the essential aspects of self-consistency, faithfulness, and active engagement in discussions.

---

**G-EVAL**

You will be given one metareview written for reviews by the committee on a paper. Your task is to rate the metareview on one metric. Please make sure you read and understand these instructions carefully. Please keep this document open while reviewing, and refer to it as needed.

Evaluation Criteria: Quality of Metareview (1-5) - the collective quality of all sentences. We align this dimension with the DUC quality question of structure and coherence whereby the metareview should be well-structured and well-organized. The metareview should always discuss the disadvantages and advantages of a paper and have a clear scope of the accept/reject decision. The metareview should have concrete evidence from the papers reviews and concrete comments as well.

Evaluation Steps:
1. Read the reviews carefully and identify the main topic and key points.
2. Read the metareview and compare it to the reviews. Check if the metareview covers the main topic, discusses advantages and disadvantages, if the most important advantages and disadvantages discussed in the above meta-review, if the most important advantages and disadvantages discussed in the above meta-review, if the most important consensus and controversy discussed in the above meta-review, if the above meta-review contradicting reviewers' comments, if the above meta-review supporting the acceptance/rejection decision, and if it presents them in a clear and logical order.
3. Assign a score for the quality of the meta-review on a scale of 1 to 5, where 1 is the lowest and 5 is the highest based on the Evaluation Criteria.

Source Text: {Reviews}  Metareview: {Meta-review}  Evaluation Form (scores ONLY): - Quality of metareview :

**Likert scale scoring with ChatGPT**

Imagine you are a human annotator now. You will evaluate the quality of metareviews written for a conference by giving a mean value from 1 to 5 and no other explanation. Please follow these steps:
1. Carefully read the reviews, and be aware of the information it contains.
2. Read the proposed metareview.
3. Rate the summary on three dimensions: 'Discussion Involvement', 'Opinion Faithfulness' and 'Decision Consistency'. You should rate on a scale from 1 (worst) to 5 (best) and give me an average of these scores over all aspects from 1 to 5 calculated by the mean of all aspects.

Definitions are as follows:
(1) Discussion Involvement: Whether the meta-review discusses the paper's strengths and weaknesses, as well as agreements and disagreements among reviewers,
(2) Opinion Faithfulness: Whether the meta-review contradicts reviewers' comments,
(3) Decision Consistency: Whether the meta-review accurately reflects the final decisions.
Only generate the mean rating as a number on the likert scale, nothing else.

---

Figure 4: We customize the prompts in G-EVAL and GPTLikert for evaluating meta-review generation to assess discussion involvement, opinion faithfulness, and decision consistency.

---

constructed instructions for GPT models to follow, yielding a rating on the Likert scale ranging from 1 to 5. Likert scoring with ChatGPT (GPTLikert), a human-like evaluation method introduced by [Gao *et al.*, 2023], follows a similar evaluation protocol, outperforming many standard metrics on text summarization as measured by human correlation. We are the first to adapt these methods to meta-review generation by modifying the prompts as shown in Figure 4. The combination of standard metrics like ROUGE-L and BERTScore with LLM-based metrics such as G-EVAL and GPTLikert ensures a comprehensive evaluation, capturing nuances that traditional metrics may overlook. This multifaceted approach

not only adheres to current evaluation methodologies, but also enhances them by introducing metrics that demonstrate a high correlation with human annotations.

# 7 Experiments

---

**Human**
The paper unanimously receives positive rates thanks to strong motivations and interesting results. As the reviews show satisfaction on the authors' feedback, the final draft needs to respect it accordingly, for example, about the limitations of this research.

---

**Vanilla**
This paper could have benefited from a clearer and more concise presentation of the main findings. It is sometimes difficult to follow the paper's chain of thought, and the authors could have presented their results in a more accessible way. The experiments are limited to a small set of environments, and it would be interesting to see how the results generalize to other tasks. Additionally, it would be useful to have a deeper analysis of why iGPT does not perform well, as this could help inform future research on how to improve image-based pretraining.

---

**CGI$^2$**
After carefully considering the reviews provided, there is consensus that the paper presents an important and thought-provoking analysis of representation dynamics between pretraining and finetuning in the context of reinforcement learning. Both reviewers agree that the paper addresses an important question and presents insightful experiments. One reviewer highlights the experiments as a strong point in the paper, while the other reviewer raises a concern about the paper's tendency to present hypotheses without exploring them further.
Overall, the strengths of the paper outweigh its weaknesses, and it has the potential to make a significant contribution to the field of reinforcement learning. Therefore, I recommend accepting the paper with a minor revision to address the concern raised by the reviewer. Specifically, the paper should explore the hypotheses presented in more depth to provide a more comprehensive analysis. If the authors can address this concern, the paper can make a valuable contribution to the field of reinforcement learning. In summary, I recommend accepting the paper with a minor revision.

---

**Initial run**
Based on the reviews, I recommend accepting the paper. The paper addresses an important and under-explored research question and presents a detailed and thought-provoking analysis of the pretraining-finetuning paradigm for Transformer models in the context of reinforcement learning. The paper also identifies relevant techniques to perform analysis of representation similarity and training dynamics. While there are some weaknesses pointed out, such as the need for further exploration of some hypotheses and occasional difficulty in following the writing and presentation, the strengths of the paper outweigh these concerns. Therefore, I recommend accepting the paper with minor revisions to address the weaknesses pointed out by the reviewers.

---

Figure 5: We show the meta-reviews from human, vanilla, CGI$^2$, and CGI$^2$ without iterative runs for the same paper. The yellow background indicates hallucinated content. The green background indicates redundant content.

## 7.1 Baselines

We compare our proposed CGI$^2$ method with methods of different paradigms. Results in Table 3 are averaged across three random runs.

**Abstractive Methods.** PlanSum [Amplayo *et al.*, 2021b] uses a Condense-Abstract Framework, where reviews are condensed and used as input to an abstractive summarization model. OpinionDigest [Suhara *et al.*, 2020] extracts opinions from input reviews and trains a seq2seq model that generates a summary from this set of opinions. MeanSum [Chu and Liu, 2019] is an unsupervised multi-document abstractive summarizer that minimizes a combination of reconstruction and vector similarity losses. LED [Beltagy *et al.*, 2020] is a Longformer [Beltagy *et al.*, 2020] variant supporting long document generative sequence-to-sequence tasks.

**Extractive Methods.** LexRank [Erkan and Radev, 2004] is an unsupervised extractive summarization method that selects sentences based on centrality scores calculated with graph-based sentence similarity. MemSum [Gu *et al.*, 2022] models extractive summarization as a multi-step episodic Markov Decision Process of scoring and selecting sentences.

**Prompting Methods.** All prompting methods are initiated with the GPT-3.5-turbo model with a temperature of 0.7. 3Sent [Goyal *et al.*, 2022] applies a simple prompt "Summary of document in 3 sentences". TCG [Bhaskar *et al.*, 2022] explores a four-step generation pipeline involving topic classification, sentence grouping by topic, generating chunk-wise summary, and generating the final summary. We also explore In Context Learning (ICL) [Brown *et al.*, 2020], where a highly rated meta-review alongside the reviews is given as part of the model's prompt. This meta-review is manually picked based on adherence to the previously defined checklist, and is chosen for its fulfillment of the criteria that define a high-quality meta-review. Vanilla uses "Generate a metareview" as the prompt. InstructPrompt provides more detailed step by step instructions and specifies the criteria for writing a metareview.

## 7.2 Automatic Evaluation

Higher standard metric scores indicate better summarization, but not necessarily better opinion summarization. ROUGE-L, BERTScore, SummaC, and DiscoScore do not consider the multifaceted nature of meta-review, which goes beyond summarization. Our method performs near average in BERTScore and SummaC, and the highest in ROUGE-L and DiscoScore amongst the prompting methods. Compared to extractive and abstractive methods, our method achieves lower scores as some metrics measure semantic similarity which a high-quality measure review with its variablility may not score well in. Additionally due to the multifaceted nature of opinion summarization, reference-based metrics such as Rouge-L can be biased towards the reference, thus the elevated scores of the summarization methods.

Evaluators like G-Eval and GPTLikert favor specific dimensions given in their prompts. Our method shows promising results in both G-Eval and GPTLikert due to the carefully constructed and revised prompts. Most prompting methods also outperform extractive and abstractive methods.

Human meta-reviews in the dataset scored among the lowest in all categories, signifying the unreliability of some human-written meta-reviews and the need for an automatic, or auxiliary, writing process. When compared by semantic similarity, extractive methods outperform both abstractive and prompting methods with the exception of Plansum. This is due to the nature of content planning in Plansum which is central to the task of meta-review generation.

| Models | ROUGE-L | BERTScore | FactCC | SummaC | DiscoScore | G-EVAL | GPTLikert |
|---|---|---|---|---|---|---|---|
| Human | - | - | 0.538 | 0.368 | 0.740 | 0.731 | 0.607 |
| *Abstractive Methods* | | | | | | | |
| PlanSum | **0.465** | 0.785 | 0.608 | 0.533 | 0.911 | 0.731 | 0.608 |
| OpinionDigest | 0.124 | 0.838 | 0.612 | 0.575 | 0.862 | 0.762 | 0.618 |
| MeanSum | 0.132 | 0.827 | 0.559 | 0.464 | 0.900 | 0.767 | 0.622 |
| LED | 0.161 | 0.846 | 0.618 | 0.785 | 0.958 | 0.731 | 0.624 |
| LED-finetuned | 0.221 | 0.853 | 0.634 | 0.795 | 0.961 | 0.751 | 0.649 |
| *Extractive Methods* | | | | | | | |
| LexRank | 0.433 | **0.881** | **0.729** | **0.937** | **1.256** | 0.726 | 0.656 |
| MemSum | 0.337 | 0.827 | 0.683 | 0.825 | 0.989 | 0.711 | 0.628 |
| *Prompting Methods* | | | | | | | |
| Vanilla | 0.174 | 0.817 | 0.498 | 0.423 | 0.808 | 0.752 | 0.626 |
| 3Sent | 0.109 | 0.783 | 0.562 | 0.503 | 0.667 | 0.758 | 0.661 |
| InstructPrompt | 0.208 | 0.823 | 0.543 | 0.449 | 0.862 | 0.751 | 0.646 |
| TCG | 0.189 | 0.847 | 0.544 | 0.466 | 0.895 | 0.761 | 0.632 |
| ICL | 0.192 | 0.847 | 0.578 | 0.470 | 0.871 | 0.756 | 0.612 |
| **$CGI^2$** (ours) | 0.199 | 0.836 | 0.559 | 0.320 | 0.906 | **0.770** | **0.687** |
| $CGI^2$ w/o Iterative Runs | 0.118 | 0.830 | 0.536 | 0.332 | 0.849 | 0.732 | 0.629 |

Table 3: ROUGE-L and BERTScore assess semantic similarity with reference text. FactCC and SummaC detect factual consistency. DiscoScore measures coherence. G-EVAL and GPTLikert are GPT-based comprehensive evaluation measures for discussion involvement, opinion faithfulness, and decision consistency.

| Model | Informativeness | Soundness | Self-consistency | Faithfulness |
|---|---|---|---|---|
| Human | 0.71 | 0.68 | 0.67 | - |
| LED-finetuned | 0.56 | 0.46 | 0.21 | 0.73 |
| LexRank | 0.87 | 0.94 | 0.16 | - |
| **$CGI^2$** (ours) | **0.98** | **0.92** | **0.84** | **0.79** |
| $CGI^2$ w/o Iterative Runs | 0.97 | 0.76 | 0.48 | 0.74 |

Table 4: Human annotation results on meta-reviews for 50 challenging papers from the test set.

## 7.3 Human Evaluation

We conduct a human annotation on 50 challenging papers from the test set which have average review scores on the borderline of acceptance. Five anonymized outputs from Human, LED-finetuned, LexRank, $CGI^2$, and $CGI^2$ without iterative runs, are shown to three annotators. Annotators are asked to provide binary labels for informativeness, soundness, self-consistency, and faithfulness for each meta-review. Informativeness measures whether the meta-review involves a discussion of both strengths and weaknesses. Soundness examines whether the meta-review provides evidence to support the discussed strengths and weaknesses. Decision consistency indicates whether the recommendation decision is clearly written and consistent with the comments in the meta-review. Faithfulness evaluates whether the meta-review contains hallucinations. We assume Human and the extractive LexRank framework have perfectly faithful summaries.

Results shown in Table 4 validate the effectiveness of our proposed method. The extractive method (LexRank) is easily biased toward one reviewer and involves no discussion or decision, but generates no hallucinations by construction. The abstractive method (LED-finetuned) learns to copy the sentences in the input and form a short meta-review with little discussion, sometimes hallucinating or generating repetitive outputs. Our prompting-based method exhibits less hallucination due to the evidence requirements in our

prompts. Compared to human-written meta-reviews, all automatic methods are less capable of generating in-depth analyses, a deficiency which calls for knowledge enhancement that happens a LLM enhanced with reviews.

We also observe that hallucinations in LLMs are more likely to happen when summarizing consensuses and controversies, which require information from the paper itself. By contrast, the abstractive methods' hallucinations were are more likely to be general comments, whereas extractive methods tend to misrepresent the context by selecting irrelevant or less important sections. Despite our method's improvements in this area, hallucination detection for scientific opinion summarization remains an open problem.

## 7.4 Case Study

Figure 5 presents the meta-reviews from human, vanilla, $CGI^2$, and $CGI^2$ without iterative runs for a random paper[3].

We make the following general observations: (1) The hallucination problem is alleviated in $CGI^2$ as the model is constantly asked for evidence. (2) $CGI^2$'s summary sentences are redundant. (3) The vanilla prompting baseline does not make recommendations and involve discussion, as the model fails to fully understand the complex task requirement. (4) Iterative refinement sometimes improves the concreteness of

---

[3]https://openreview.net/forum?id=9GXoMs__ckJ

opinion discussion. However, there are two problems with iterative refinements. First, suggestions provided by the large language model are usually generic and less useful for further refinement. Second, more self-refinement iterations cause the model to forget the initial instructions for opinion extraction and discussion.

## 8 Conclusions and Future Work

In this paper, we introduced the task of scientific opinion summarization, in which research paper reviews are synthesized into meta-reviews. To facilitate this task, we introduce the ORSUM dataset, an evaluation framework, and an approach that we call Checklist-Guided Iterative Introspection. We conduct an empirical analysis of methods from different paradigms, concluding that human-written summaries do not always satisfy the criteria of an ideal meta-review, and that the combination of task decomposition and iterative self-refinement shows promise in on this task.

Direct extensions of this work include the incorporation of author rebuttals into the input data to enhance the model's ability to generate more balanced meta-reviews, and introducing an effective and efficient hallucination detection tool for scientific opinion summarization.

## Limitations

This work on scientific opinion summarization has limitations in terms of data scope and task configuration. As the dataset is collected from OpenReview, the majority of meta-reviews are in Machine Learning, and many papers have been accepted. Conclusions drawn from this data distribution might not be applicable to datasets in other domains. Furthermore, to simplify the task setting, author rebuttals have not been included as input, which may also constrain the extent of discussion involvement in generating meta-reviews. section*Ethics Statement

We acknowledge the following potential ethical concerns that may arise. First, the meta-reviews generated by LLMs may contain hallucinations, which may lead to misunderstandings of the original research paper or reviewers' opinions. Therefore, users should be cautious when using system-generated meta-reviews for recommendation decisions. Second, the use of black-box LLMs for meta-review generation may raise concerns about the transparency of the decision process. Though our method improves explainability by prompting an LLM to provide supporting evidence for the recommendation decision, the evidence may not perfectly reflect the decision-making process. Third, the dataset used in this study mainly focuses on machine learning papers, which could introduce biases to the recommendation decisions. Hence, it is critical to consider these biases when applying our method to generate meta-reviews for research papers in other domains.

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
