# OpenReview forum: "Scientific Opinion Summarization: Paper Meta-review Generation Dataset, Methods, and Evaluation"
_ijcai.org/IJCAI/2024/Workshop/AI4Research — AI4Research 2024_

### Official Review · Reviewer_T1qW · 2024-06-01

**Rating:** 6
**Confidence:** 4

**Review:**

**Summary:**
Meta-review generation can be seen as a form of opinion summarization, except that in paper meta-review, we not only gather opinions but also make recommendations/decisions. Authors define this new task as Scientific Opinion Summarization. The main contributions are:
1) dataset: ORSUM, which collects open-sourced papers and meta-reviews from OpenReview. This dataset covers 15,062 meta-reviews and 57,536 reviews from 47 different conference venues.
2) a system: Checklist-guided Iterative Introspection (CGI^2) to solve the Scientific Opinion Summarization task. Given the title, abstract, and individual reviews, there are two steps in CGI^2 to obtain a meta-review:
step 1: extract opinions --> summarize pros and cons --> summarize consensus and controversy --> write a Meta-review
step 2: Authors prepare a predefined list of questions to ask the LLM to evaluate the review. Given the generated review and a question from the list, the LLM will iterate using the self-evaluation to keep improving the Meta-review.

**Strength:**
1. Comprehensive evaluation. There are popular Standard Metrics for relevance, factual consistency, and semantic coherence and LLM-based Metrics for discussion involvement, opinion faithfulness, and decision consistency.
2. The writing is clear, and the experimental design is explained well with details.

**Weakness:**
1. Even though I agree that this meta-review generation task is closely correlated with opinion summarization, I also suggest you compare it with datasets/systems designed explicitly for meta-review, such as the MRed dataset you mentioned.
2. For many venues, such as ICLR, the reviews can be thousands of words long. It seems that most information would be lost if each review is truncated to 300 tokens.
3. LLM-based evaluation has many advantages that traditional methods do not, such as better semantic understanding and reasoning abilities. I agree that this is a must-have piece for such a task. However, your checklist is addressing almost exactly the questions you will be evaluating in G-EVAL or GPTLikert. It might seem like cheating when you use the same model to produce and evaluate itself with a similar set of questions. So, the question here might be how subjective it can be.
4. It looks like all experiments are based on GPT-3.5-turbo. As a work that completely relies on GPT, I don't see a reason why GPT-4 is not considered. Given that GPT-4 is coming out with a much cheaper price, at least a subset is recommended.

---

### Official Review · Reviewer_LEmm · 2024-06-01
**Paper Review**

**Rating:** 7
**Confidence:** 4

**Review:**

In this paper, the authors first introduce a dataset ORSUM with reviews and meta-reviews collected from 47 conferences for the evaluation of the meta-review generation task. Then, the authors propose a Checklist-guided Iterative Introspection approach to break the complex summarization into stages and iteratively run a self-refine algorithm with LLMs. Results show that human written summary may not satisfy all requirements and the proposed approach achieves a good score on human evaluation.

Strength:

1. Meta-review generation is an interesting topic for it can help ACs to save labor and make more fair decisions.

2. The dataset ORSUM covers a wide range of papers, consisting of 47 conferences. It potentially could contribute to facilitating the research on review generation tasks.

3. The paper is clear and well-motivated. It is relatively easy to read and follow.

Weakness:

1. Missing some related work and comparison. For instance
- Miao Li, Eduard Hovy, Jey Lau. Summarizing Multiple Documents with Conversational Structure for Meta-Review Generation
- Asheesh Kumar, Tirthankar Ghosal, Saprativa Bhattacharjee, Asif Ekbal. Towards automated meta-review generation via an NLP/ML pipeline in different stages of the scholarly peer review process


2. Factuality issue. The proposed approach shows a lower FactCC and SummaCC score than traditional neural abstractive summarization. Although human evaluation shows a higher score, it is still unknown how severe the hallucination is. At least some analysis is needed to show the proposed approach is reliable.

3. Missing implementation details. Missing the details of dataset construction, such as what are the 47 conferences? Are they all AI papers? What is the average number of reviews for each paper? Missing the details of models. Such as the backbone of the proposed method, the backbone of G-Eval, etc.

4. Fine-grained evaluation. Compared with an automatic evaluation of the whole summary such as ROUGE, it is better to use metrics that evaluate some part of the source (Yuan et al.). For instance, is the decision of the generated summary is correct? is the summary factually correct? Is the Weakness vs. Strength fair and comprehensive?

- Weizhe Yuan, Pengfei Liu, Graham Neubig. Can We Automate Scientific Reviewing?

Presentation:
I suggest increasing the font size of figures 1-4 and changing it to black color.

---

### Decision · Program_Chairs · 2024-06-03

Accept